# Entropy Generation and Exergy Analysis of Premixed Fuel-Air Combustion in Micro Porous Media Burner

**DOI:** 10.3390/e22101104

**Published:** 2020-09-30

**Authors:** N. C. Ismail, M. Z. Abdullah, N. M. Mazlan, K. F. Mustafa

**Affiliations:** 1School of Mechanical Engineering, Universiti Sains Malaysia, Engineering Campus, Nibong Tebal 14300, Penang, Malaysia; nazmi5075@yahoo.com (N.C.I.); mekhairil@usm.my (K.F.M.); 2School of Aerospace Engineering, Universiti Sains Malaysia, Engineering Campus, Nibong Tebal 14300, Penang, Malaysia; nmusfirah@usm.my

**Keywords:** porous micro-burner, exergy efficiency, entropy generation, rich fuel combustion, surface flame

## Abstract

The performance of porous media micro-burners plays an important role in determining thermal efficiency and improving our daily life. Nowadays, a lot of scholars are actively involved in this research area and ongoing studies are still being carried out due to the burners’ excellent performance. The exergy efficiency and entropy generation of a porous media burner are strongly dependent on the characteristics of the flame and its thermal behavior. In this study, a single-layer and double-layer porous media form were constructed to investigate the effects of various types of porous foam arrangement in a cylindrical burner. The burner was operated using premixed butane-air combustion with an inner diameter of 23 mm and a length of 100 mm. The experiments were carried out in rich fuel conditions with an equivalence ratio, φ ranging from 1.3 to 2.0. The results showed significant improvement in the thermal and exergy efficiency with an increase in the equivalence ratio in a double-layer compared with a single-layer. The peak temperature recorded was 945.21 °C at φ = 1.3 for a porcelain single-layer, and the highest exergy efficiency was 83.47% at φ = 2.0 for an alumina-porcelain double-layer burner. It was also found that the average temperature of the burner wall decreased with an increase in the equivalence ratios for PMB2 and PMB4, whereas the average wall temperature for PMB3 was largely unaffected by the equivalence ratios. The total entropy generation rate reached the highest value at φ = 2.0 for all PMB configurations, and the highest percentage increase for total entropy generation rate was 46.09% for PMB1. The exergy efficiency for all burners was approximately similar with the highest exergy efficiency achieved by PMB4 (17.65%). In addition, the length and location of the flame with thermal distribution was significantly affected by the equivalence ratio between the single-layer and double-layer porous material. Overall, a double-layer porous media burner showed the best performance calculated based on the second law of thermodynamics when compared with other configurations, and it is ideal for domestic application.

## 1. Introduction

Applications of porous media burners have been a major concern to researchers worldwide over the last few decades, but they are still used in the real world due to their immense benefits, such as high thermal performance and low emissions. Domestic use and commercial production are the most common fields in everyday use which consume significant quantities of fossil fuel [1,2]. Therefore, in order to reduce usage and pollution, porous media burners (PMBs) are developed because they have more advantages and benefits. PMBs can increase energy efficiency and provide economic benefits using premixed air-fuel burners with a double-layer structure [3]. The fundamental principle is based on the continuous flow and reaction between the fuel and air mixture in the porous matrix. There are numerous applications of PMBs in household heating appliances, reformers and heat exchanger systems [4]. There are also reported applications in water heating [5], glass and chemical processing, coating and paint drying [6]. In addition, the potential of PMBs can also be found in landfill combustion [7] and pressure cook-stove [8].

In order to improve the energy conversion efficiency, many works or methods have been studied and tested such as the size of burners [9], porous media materials [10,11], layer of pores [12,13], and the type of fuel-air consumptions [14]. Many authors have done novel studies in determining the effectiveness of non-premixed [15,16] and premixed combustions [17,18]. Premixed combustion increases the performance when operating with flame control and low emissions and decreases the residence time of the gas mixture. Gao et al. [19] experimentally tested a single-layer burner packed with 3 mm diameter pellet compared with a double-layer burner packed with various alumina pellet diameters (6, 8, 10, and 13 mm). They found that the flame stability limit could be extended in the double-layer burner relative to the single-layer burner. In addition, the CO emission of a double-layer burner was much lower than that of a single-layer burner at low flame velocity. Hayashi et al. [20] also studied double-layer burners using alumina materials but in the foam shapes with varying layer thickness.

Detailed studies about combustion characteristics, which are related to the second law of thermodynamics, received greater attention over the last decade. Researchers tried to find alternative ways to reduce heat loss during combustion due to an irreversibility process. Several approaches have been researched and tested in order to improve energy consumption efficiency. Therefore, the irreversible energy loss in combustion of fuel cannot be ignored. In order to improve thermal performance, Nadimi and Jafarmadar [21] used a micro-fin inside the micro-combustor. Their results showed that the micro-fin has a positive effect on the energy and exergy efficiency, as well as on the external wall temperatures. In addition, Jiang et al. [22] used baffles in a micro-combustor for computational calculations and analyzed in terms of exergy efficiency and entropy distribution using various equivalence ratios, mass flow rate, and baffles height. They reported that the exergy efficiency of the exercised was low at the closest stoichiometric equivalence ratio. Jiaqiang et al. [23] investigated the effect of inlet pressure between 0 MPa and 0.1 MPa on premixed combustion in a micro-cylindrical combustion with a step. The results suggested that high inlet pressure has the potential to improve the energy conversion and exergy efficiency at an inlet pressure of 0.1 MPa.

Based on the second law of thermodynamics, the effect of irreversibility must be minimized in order to increase the combustion efficiency of a system in the environment. Therefore, the generation rate of the entropy in the system needs to be overcome. Exergy destructive, exergy loss, and exergy efficiency have been examined by Rana et al. [24,25] in a cylindrical micro combustor using heat recirculation premixed flame. It was stated that the gas phase heat conduction and chemical reactions in entropy generations were key sources of energy destruction. Moreover, exergy loss at the outer wall caused about 5–7 percent of inflow exergy while exergy destruction from the heat recirculation and combustion caused about 40–45 percent of inflow exergy.

Peng et al. [26] investigated the generation of entropy in premixed hydrogen/air combustion and studied the effect of inlet mass flow and equivalence ratio. The findings revealed that the generation rate of the entropy increased with an increase in the hydrogen mass flow rate. In addition, the generation rate of entropy for a narrow inlet micro tube tended to be lower than the straight tube with no inlet. Arjmandi and Amani [27] focused on numerical simulation for entropy generation and thermodynamic optimization in the combustion chamber of a gas turbine. Their studies concluded that the entropy generation rate was heavily influenced by the chemical reactions and heat transfer in the combustion chamber. More recent works on entropy generation for various shapes of cavities were also reported by Armaghani et al. [28], Mansour et al. [29], and Rashad et al. [30]. Armaghani et al. [28] for example, conducted a numerical investigation in an I-shape porous media to determine the heat transfer and the entropy generation. The work was primarily based on the earlier investigation of Mansour et al. [29] on a C-shape cavity to observe the effects of heat transfer on the rate of entropy generation. The rate of entropy generation was also extended to a U-shape cavity, as elucidated by Rashad et al. [30]. Table 1 presents some of the most recent studies and findings on the effect of porous media materials on entropy generation, S_gen_, in combustion systems.

The equivalence ratio, φ in premixed combustion can also overcome the exergy loss and efficiency during the combustion process. The effect of fuels, equivalence ratio, and inlet temperature for both hydrogen and methane premixed flames and diffusion flames are key factors in local entropy generation, as claimed by Nishida et al. [41]. They found that the chemical reactions were major contributing factors, which controlled the entropy generation and exergy loss. In order to reduce the volumetric local entropy generation rate, Yapici et al. [42] conducted a numerical analysis on the effect of equivalence ratio and oxygen fraction in a turbulent premixed-air burner. The results suggested that an increase in the equivalence ratio significantly reduced the reaction rate of entropy generation.

Although the entropy generation and exergy efficiency performance of premixed fuel combustion in micro-cylinder burners has been extensively investigated, no research has been reported on the effect of porous media layers in micro-cylinder burners on the entropy and exergy performance using premixed butane-air combustion. It is well known that the configuration of a porous media in a micro-cylinder is another method for improving the efficiency of exergy and reducing entropy generation in combustion. Flame stability, temperature variations, energy destroyed, and emission pollutions have been studied as additional findings in these researches. The aim of the present work is to investigate the performance of micro burners for different porous media configurations using premixed fuel-air combustion by evaluating entropy generation and exergy efficiency.

## 2. Methodology

### 2.1. Experimental Setup and Procedures

The layout of the experiments setup is shown in Figure 1 using a micro-cylindrical shape with an inner diameter (ID) of 23 mm and a length of 100 mm. The burner was made of mild steel with a wall thickness of 2 mm. The PMB considered for the present work focused on both single-layer and double-layer porous media. While PMB1 (inlet nozzle, ID = 11 mm without the burner housing) was configured as a reference measurements in this study. The burner, which has one layer, has porous media placed on top, where the reaction happens, which is named the reaction zone. Meanwhile, in the burner which has two-layers, alumina foam acts as reaction zone (top) and porcelain foam acts as preheat zone (bottom). The material and setup used in this study are similar to the work conducted by Janvekar et al. [43], and Table 2 shows the properties of the porous media materials used in this investigation.

The foam thicknesses of alumina and porcelain were fixed at 15 and 10 mm, respectively. The fuel and air flow rates were determined using a digital flow meter with a control valve (Model: Red-y compact series; Vögtlin Instruments) in liters/min for mass flow rate measurements. The experiments were performed using compressed butane cartridge (Model: MBG-230EP; Milux Sdn. Bhd., Malaysia) as a fuel, while the air pump (Model: HP4000; Atman, China) was used to provide natural air. Air and fuel were supplied using different air-fuel ratios and transported through a rubber tube to the burner. The mixing unit, reinforced with wire steel mesh and mounted just below the burner, provided excellent premixed air-fuel before entering the burner chamber.

A portable flue gas analyzer model (Model: Kane 251 flue gas analyzer; KANE International Ltd., Welwyn Garden City, UK) was used for recording carbon monoxide, CO and nitric oxide, and NO emissions in part per million (ppm) and placed it on the top of the surface flame using a smokestack collector. For each experiment performed, the equivalence ratios, φ, were calculated from the value of air and fuel flow rates, respectively, as shown in Equations (1) and (2). The specific equivalence ratios were obtained by changing the air and fuel flow rates within the range. The chemical equation for butane gas and natural air in a complete combustion is as follows:C_4_H_10_ + 6.5 (O_2_ + 3.76 N_2_) → 4 CO_2_ + 5 H_2_O + 24.44 N_2_(1)

Therefore, the calculations for the equivalence ratio, φ, are based on the values of *Air/Fuel_stoic_* obtained from Equation (1) divided by the values of *Air/Fuel_actu_* obtained from measurement of the experiment using a digital flow meter.
(2)$$\varphi \;=\;\frac{Air/Fuel_{stoic}}{Air/Fuel_{actu}}\;=\;{\left(\frac{{ṁ}_{O2+3.76N2}}{{ṁ}_{C4H10}}\right)}_{stoic}\;/\;{\left(\frac{{ṁ}_{O2+3.76N2}}{{ṁ}_{C4H10}}\right)}_{actu}$$

Temperatures were recorded using DAQ systems (Model: USB-4718; Advantech) and stored via personal computer (Model: Pavilion p6345d; Hewlett-Packard). The temperature readings were measured using eight K-type thermocouples supported by a stand. The experiments were conducted at room temperature for initial conditions and the porous media burner dimensions were shown in Figure 2. All of the thermocouples were placed at the top of porous media burner of varying heights. The function of thermocouples is to measure the surface flame temperatures. In addition to that, Figure 2 represents the configurations of the porous media used in these investigations. To investigate the average wall temperature of porous PMB configurations, the thermocouples were placed on the side of the porous media burners, and these temperatures were recorded using the same DAQ system and stored via personal computer as shown in Figure 3. The thermocouples T1, T2, and T3 were used to measure the average porous wall temperature for alumina foam, while T4 and T5 were used to measure the average porous wall temperature for porcelain foam.

The premixed butane-air starts up using external ignition electrodes to ignite the mixture at the top of the burner. Once ignited, the fuel and air were adjusted to the desired flow rate using a digital flow meter to yield the exact equivalence ratio. The data collected from the experiments were increased by 0.1 at a rich condition between equivalence ratios of 1.3 to 2.0. Then, premixed butane-air data collection and exhaust gas sampling were carried out after 30 min for each trial to maintain steady-state conditions. This procedure ensured that the burner did not experience blow-off or flashback phenomenon during combustion. The same procedure was applied to other porous media burner configurations (i.e., no porous (PMB1), alumina—1 layer (PMB2), porcelain—1 layer (PMB3), and alumina-porcelain—2 layers (PMB4)).

### 2.2. Parameter Calculations

The following equations were used to determine the efficiency of parameters based on the energy and exergy principle. The burner performance based on water heating method for efficiency purposes was calculated based on the conservation law of energy, which ensured that the energy supplied was equal to the energy produced by combustion, Q_total_. The method used to calculate these parameters was similarly done by Janvekar et al. [43] and Ismail et al. [44]
ṁ = ρ × ύ(3)
Q_total_ = ṁ × C_v_(4)
where, C_v_ is the calorific value of butane fuel. The derivations of mass flow rate, ṁ, were obtained from the density (ρ) and volumetric flow rate (ύ) of butane. The energy generated from combustion is denoted by Q_actu_.
Q_actu_ = [(M_w_C_w_ + M_c_C_p_) (50 °C − T_o_)]/t(5)
where, M_w_ and M_c_ are the masses of water and container in kilogram (kg), respectively. Standard values for specific heat of water and container are C_w_ = 4.1826 kJ/kg K and C_p_ = 0.5024 kJ/kg K, respectively. The water in the container was weighted, then, left to heat-up until the final temperature reached 50 °C from the ambient temperature, T_o_, with time, t (s) recorded.

In the combustion reactions, the basic thermodynamics principles from the second law provide direct derivation of entropy generation and exergy analysis. The energy loss from the systems can therefore be described as follows:E_loss_ = Q_total_ − Q_actu_(6)

Then, the total exergy in the combustion is
Q_exergy_ = (Q_actu/_ṁ) − T_o_ (S_max_ − S_amb_)(7)
where, S_max_ is the entropy at maximum temperature, and T_max_ and S_amb_ is the entropy at an ambient temperature, T_o_, respectively.

In this case, the total entropy generation, S_gen_, can be considered as follows [23,26]:S_gen_ = ṁ × Δs(8)
where Δs is the specific entropy different, S_max_ – S_amb._

Due to the exergy destroyed, E_des_ in a combustion process is proportional to the total entropy generation, and it is expressed as follows:E_des_ = S_gen_ × T_o_(9)
where T_o_ is the ambient temperature, 302 K.

Therefore, the exergy efficiency is defined as follows:η_exergy_ = (Q_exergy_/Q_total_) × 100%(10)

### 2.3. Analysis of Uncertainty

The error variance is also an important aspect in the measurement analysis. In this study, the errors are expected to occur in the measurements. Thus, the error analysis is essential to check the accuracy of the measured quantities. Random errors can be easily detected and analysed using statistical analysis. The flame temperature, average porous wall temperature, and emissions are the main quantities tested for error measurements. Each experimental test was carried out 3 times and taken as the average error analysis. The mean ($\bar{x}$), standard deviation ($\sigma_{x}$), standard error ($\sigma_{\bar{x}}$), and uncertainty (U_n_) were calculated and presented in Table 3. The uncertainty was calculated based on 95% confident interval from the student t_(N-1)_ table with a degree of freedom 2. It was found that the maximum uncertainty for the error measurement was ±24.33 °C (2.6%) of the maximum flame temperature.

## 3. Results and Discussions

### 3.1. Effects of Different Porous Layer on Flame Stability and Temperature Variation

The flame characteristics and thermal performance of the burner are good indicators of the suitability of the porous media burner for the premixed air-fuel combustion. A comparison research on micro-burner combustion with different porous media configurations was performed to investigate the performance and characteristics with several values of equivalence ratios. Table 4 presents the detailed mass flow rate of butane fuel and air used with corresponding values of the equivalence ratios and flame velocity. Porous media porosity and configurations are important factors that affect the flame stability and temperature distributions inside the burner. The experimental results were collected from burner size, ID = 23 mm with four different PMB configurations, namely, (1) PMB1, (2) PMB2, (3) PMB3, and (4) PMB4 and tested with equivalence ratios ranged from φ = 1.3 to 2.0.

Figure 4a,b shows the photographic image for surface flame characteristics for all PMB configurations at φ = 1.3 (highest temperature) and φ = 2.0 (lowest temperature) using a digital camera. From the observations, it can be seen that at φ = 1.3, the surface flame is stable and produces the blue flame for all PMB configurations, while at φ = 2.0, the surface flame is unstable with the flame front twisted, and it produces and orange flame for all PMB configurations. Moreover, the orange flame is almost lifted from the top of the porous burners. Mostly, at φ = 1.3, the flame is almost completely combusted because the value of φ is nearly at stoichiometric condition. This happens due to the increase in the mass flow rate—with increasing values of equivalence ratios thus affecting the velocity of the flame and chemical reaction rate in the PMB. The results show that the flame length increases from φ = 1.3 to φ = 2.0 due to an increase in the mass flow rate of fuel.

The maximum flame temperature was recorded at various values of equivalence ratios for several PMB configurations as shown in Figure 5. It is also observed that the maximum flame temperature decreases with an increase in the equivalence ratio except for PMB2 where it starts to increase at φ = 1.8. The maximum flame temperature is identified as the peak temperature among eight thermocouples placed on top of the burner at various equivalence ratios [19]. Moreover, the flame location was also identified as the position where the temperature reaches the highest temperature values as shown in Figure 4. The maximum flame temperature is observed at 945.21 °C for PMB3 with φ = 1.3, while the maximum flame temperature for PMB2 is observed at 943.17 °C for single-layer configuration. It shows that single-layer porous media produces the highest flame temperature at all equivalence ratios when compared with PMB1 and PMB2 configurations. Furthermore, PMB1 (no porous) achieved the lowest flame temperature compared with other configurations for all equivalence ratios. The lowest flame temperature recorded at φ = 2.0 for PMB1 was 820.91 °C. The results show that porous materials helped the burner to enhance heat recirculation and retentions inside the burner. Moreover, only a porous single-layer has a reaction zone that provides higher heat input into the flame compared with double-layers, which have preheat and reaction zones. This effect occurs due to greater flame stability in double-layers than in single-layer with fluctuations of flame velocity. Thus, double-layer porous media generated more internal heat recirculation in the preheat zone, and heat was delivered from the reaction zone to the preheat zone by transferring the radiation heat producing low temperature compared to the single-layer porous. Furthermore, porcelain foam serves as preheat zone with higher void volume and thus leads to the transmission of radiation heat resulting in increased heat recirculation.

In addition, as shown in Figure 5, it seemed that the flame locations shifted upward (increasing) from 9 to 44 mm and that these locations varied with the equivalence ratios. These observations show that the flame location has increased from the porous surface, from the lowest to the highest height for all PMB configurations. This is due to an increase in the fuel flow rate, which caused an increase in the equivalence ratio of enriched combustion. Moreover, the transition of flame colour from blue to orange blue was observed during this period. This phenomenon revealed significant impact on the exergy efficiency, as illustrated in Figure 10, for better energy conversion performance.

The average temperature of the burner wall is also an important parameter to determine the performance of the porous media combustion. Figure 6 presents the variation of the average temperature of the burner wall with the equivalence ratio for PMB2, PMB3, and PMB4 (porcelain foam acts as preheat layer, and alumina foam act as reaction layer). It shows a trend towards decreasing in average wall burner temperature with the increase in equivalence ratios for PMB2 and PMB4. The average wall temperature for PMB3 increases at φ = 1.5 and then starts to decrease slowly until φ = 2.0. The average porous wall temperature for PMB3 appears to be almost constant with an increase in the equivalence ratio due to the smaller pore size of porcelain of 26 ppcm. Whereas the highest average wall temperature obtained for PMB2 at φ = 1.3 is measured at 108.34 °C. However, the lowest average burner wall temperature with PMB4 (preheat zone) at φ = 1.3 is measured at 66.63 °C.

### 3.2. Effects of Different Porous Layer on Entropy Generation Rate, Energy Loss and Exergy Destroyed

The variations in the equivalence ratio, φ, of premixed fuel-air combustion have affected the total entropy generation rate as depicted in Figure 7. The results show the interactions between variation of porous media configuration in rich fuel combustion (φ = 1.3, 1.5, 1.8, and 2.0) and the total entropy generation. The total entropy generation rate increased with an increased in the equivalence ratios for all PMB configurations. It is shown that total entropy generation rate reaches the highest value at φ = 2.0 for all PMB configurations compared with other values (φ = 1.3, 1.5, and 1.8). Furthermore, the highest percentage increase for total entropy generation rate from φ = 1.3 to φ = 2.0 is 46.09% for PMB1 followed by PMB2, PMB3, and PMB4 are 45.73%, 44.18%, and 43.31%, respectively, as shown in Figure 6.

It can be seen from the increased percentage that the PMB4 has the lowest entropy generation rate, which implies that less energy is lost during the premixed combustion. Of course, the key explanation is that the double-layer system has a superior mixing, a complete chemical reaction for the reactants, and enhanced heat flow between combustion gases and burner for various equivalence ratios.

Furthermore, the variation of entropy generation rate is proportional to the decrease in temperature while entropy generation rate increases [29]. It is attributed that the entropy generation rate is mainly determined by temperature field in premixed combustion process. Therefore, the increase in fuel mass flow rate in premixed combustion will contribute to the rise of entropy generation rate.

As depicted in Figure 8, the energy analysis was carried out to measure the exhaust gas (energy loss, E_loss_) for various equivalence ratios and PMB configurations. From the results obtained, the energy losses in PMB1 and PMB2 start to increase with higher values of equivalence ratios. While PMB3 and PMB4 increased up to φ = 1.5, then dropped down at φ = 1.8, and dropped further to a minimum at φ = 2.0. Based on Figure 5, the investigation shows that the reduction in maximum flame temperature lead to an increase of energy loss during combustion except for PMB3 and PMB4 at φ = 1.8 and 2.0. In conclusion, the flame temperature plays an important role in determined the energy loss of combustion with the effect of equivalence ratios and PMB configurations.

It was shown that the porous media material is also one of the main contributions to controlling the energy efficiency of the micro-burner combustion. It could be observed that the burner with no porous media (PMB1) gives the highest energy loss at various equivalence ratios compared to the burner with porous media.

Figure 9 shows the exergy destroyed, E_des_, under various equivalence ratios for all porous media burner configurations. It can be shown that the exergy destroyed increased with an increase in equivalence ratios for all porous media burner configurations and the peak exergy destroyed obtained at φ = 2.0. The percentage difference between the lowest and the highest exergy destroyed are 46.09% (PMB1), 45.73% (PMB2), 44.18% (PMB3), and 43.31% (PMB4). Hence, energy loss and exergy destroyed is the main key in determining the exergy efficiency in PMB with various equivalence ratios. To achieve better energy conversion efficiency with higher exergy performance, both need to be minimized.

### 3.3. Effects of Different Porous Layer on Exergy Efficiency

From Equation (10), Figure 10 shows the exergy efficiency, ƞ_exergy_, obtained for various porous media configurations at different values of equivalence ratios. The exergy efficiency for PMB2, PMB3, and PMB4 increased with an increase in the premixed butane-air equivalence ratios except for PMB1. Besides, the results also show the performance of PMB4 is higher than that of other burners for all equivalence ratios. The highest exergy efficiency is 83.47% for PMB4 while the lowest is 47.40% for PMB1 at φ = 2.0.

The porous media burner configuration (PMB1) with no porous layer has descent percentage of exergy efficiency about 16.07% from φ = 1.3 to 2.0. However, for PMB2, PMB3, and PMB4, configurations have an increased percentage of exergy efficiency, about 12.54%, 14.87%, and 17.65%, respectively. This is due to the energy released from the exhaust gas (E_loss_) which mainly contributed towards the decrease of exergy efficiency. Increases in the fuel mass flow rates in premixed combustions have significant impact on the exergy efficiency performance. It can be concluded that the higher energy released via exhaust gas (E_loss_), the lower the exergy efficiency obtained [23].

Overall, it can be summarized that porous materials and configurations affect the characteristics outcome of entropy generation rate and exergy efficiency. For better energy usage, PMB4 micro burner has the best performance for all tested equivalence ratios and provides better energy conversion with higher exergy efficiency during combustion.

### 3.4. Effects of Different Porous Layer on Pollution Emission

The pollution emission of CO and NO gases with various equivalence ratios were successfully measured by using Kane 251 flue gas analyzer. Figure 11 and Figure 12 show the effect of porous media configurations on CO and NO emissions in PMB. Since the maximum CO and NO emissions concentrations are below 100 ppm, they are within reasonable limits and do not constitute a hazardous health problem.

As shown in Figure 11, the CO emissions for fuel-air premixed combustion increased rapidly for PMB2 with an increase in the equivalence ratio compared to PMB1 and PMB3. Meanwhile, the CO emission for PMB4 increased initially, and then, it started to decrease at φ = 1.5. CO emissions increase for PMB2 is attributed to the difference in the porosity size, which has a bigger size compared to PMB3. In addition, CO emission increase because the CO was not completely converted into CO_2_ due to the insufficient amount of air during combustion. As a result, higher temperature was observed at φ = 1.3 for all configurations, possibly due to the completeness of combustion because CO emissions are low. Thus, the pore size of PMB plays an important role in CO emissions production. Similar results obtained by Yu et al. [10] stated that the changes in porous media porosity could reduce CO emissions in domestic/compact household burner systems.

As shown in Figure 12, the NO emissions for all PMB configurations of fuel-air premixed combustion increase when the equivalence ratio used increases. The maximum NO emissions obtained using PMB1 configuration at φ = 2.0 is 22 ppm followed by PMB3, PMB4, and PMB2. While the lowest NO emissions obtained using PMB2 configuration at φ = 1.3 and φ = 1.5 is 3 ppm. In general, NO emissions produced when there is an excess air during combustion that leads to large amount of NO emissions. Generally, all the results obtained show that NO emissions produced are below 25 ppm for all PMB configurations. Furthermore, an increase in the NO emissions with respect to the change in equivalence ratio was observed as the fuel mass flow rate increases with an increase in the flame velocity. In this research, higher NO emission obtained for PMB1 with no porous media installed inside and higher porosity in PMB3 configuration lead to slow mixture velocity before reaching the combustion inside reaction zone (poor preheating). However, due to the differences in porous media porosity, PMB2 shows the lowest NO emissions for all equivalence ratios. In other words, the biggest pore size (with low porosity percentage) increased the exit mixture velocity, thus residence time of the reaction flow decreases, which leads to the reduction of NO emissions production (effect from increase in mass flow rate). However, PMB4 configuration is still within moderate value of NO emissions compared to other PMB configurations in dealing with pollution emissions.

## 4. Conclusions

The exergy analysis and entropy generation of premixed fuel-air combustion were successfully explored experimentally using single- and double-layer porous media burners. This research introduces a simple method of analysis to study thermodynamic irreversibility in micro porous media combustion. The effect of porous media configurations in micro burners with ID = 23 mm was investigated with various equivalence ratios. The results indicate that:The effect of porous media configurations and equivalence ratios have significant impact on the temperature distributions, entropy generation, and exergy efficiency for surface flame combustion in an open system. An increase in the equivalence ratios contributed to an increase in the exergy efficiency.The highest exergy efficiency is 83.47% for PMB4 at φ = 2.0, while the lowest exergy efficiency is 47.40% for PMB1 using a similar equivalence ratio.The flame temperature in porous media combustion also plays an important role in exergy efficiency and entropy generation. Results obtained indicate exergy efficiency and entropy generation increase proportionally with increase in equivalence ratio due to decrease in flame temperature.Based on energy conversion performance, the best micro-burner system is the PMB4 configuration because it has the highest exergy efficiency.This study proved the advantages of utilizing porous media materials and configurations to improve the performance of micro-burners in terms of thermodynamic irreversibility for fuel-air premixed combustion.

The analysis performed in this study showed that the possibility of manipulating the dimensions of the PMBs to be more compact is feasible and realistic. In addition, the intended operating variables of the PMBs can also be extended to the rich mixtures, allowing greater increase in the exergy efficiency. The utilization of a double-layer porous media has an added benefit based on the energy conversion efficiency. To conclude, the development of PMBs is highly beneficial because significant improvement in efficiency and emissions can be attained compared with conventional modes of combustion.

## Figures and Tables

**Figure 1 entropy-22-01104-f001:**
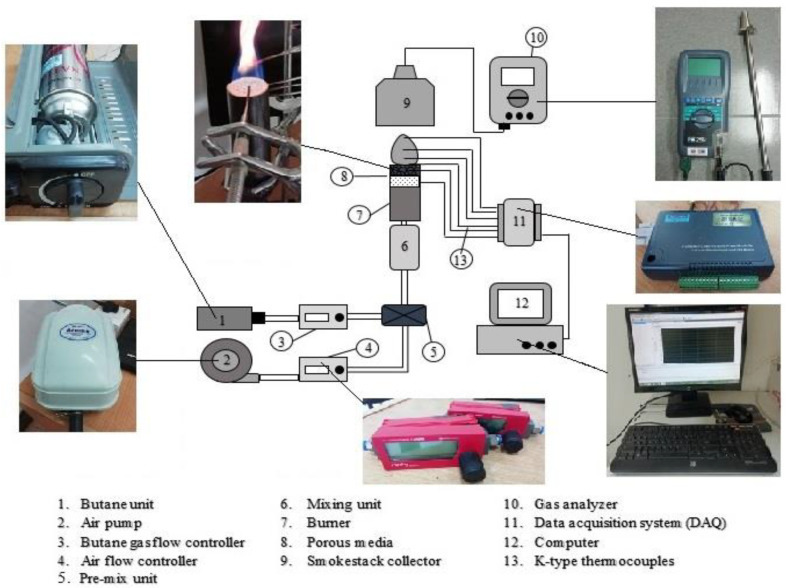
Schematic diagram of the experimental set-up for the porous media combustion system.

**Figure 2 entropy-22-01104-f002:**
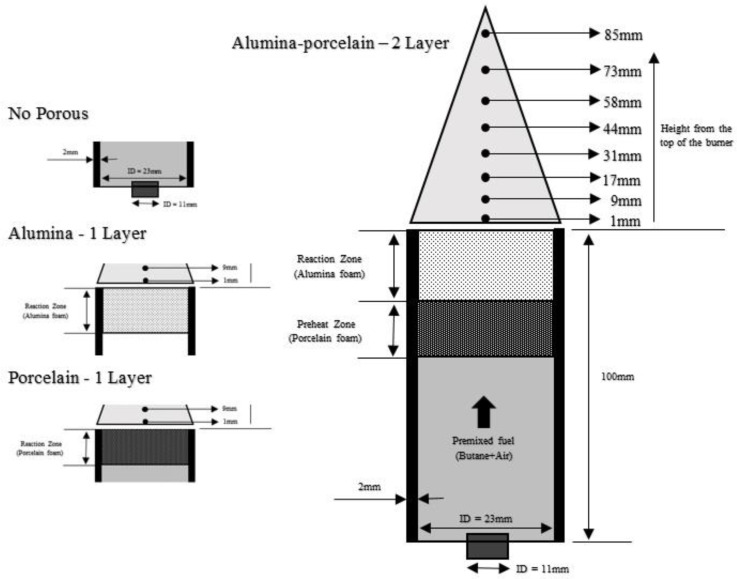
Schematic of PMB configurations and thermocouples height position measurement.

**Figure 3 entropy-22-01104-f003:**
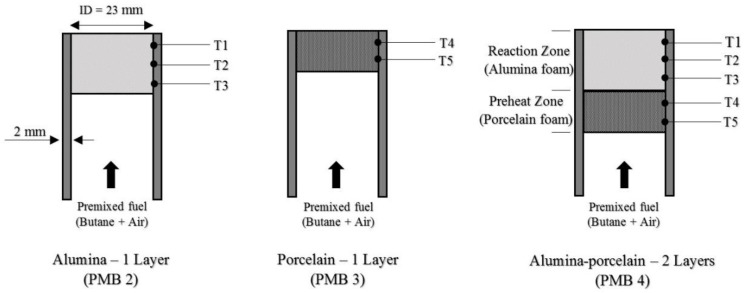
Schematic of PMB configurations with thermocouples placement for measure average temperature porous wall, T_w_.

**Figure 4 entropy-22-01104-f004:**
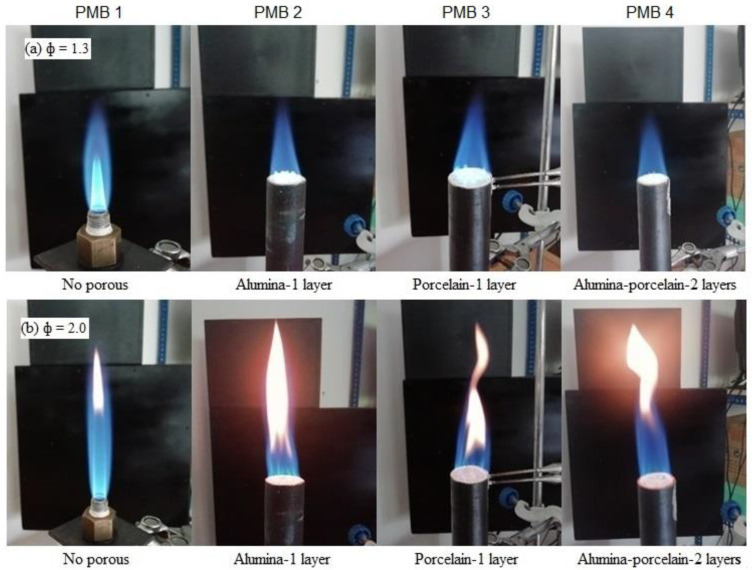
Photographs of surface flame characteristics with various PMB configurations; (**a**) φ = 1.3, (**b**) φ = 2.0. (Model: J1 digital camera; Nikon Corp., Tokyo, Japan).

**Figure 5 entropy-22-01104-f005:**
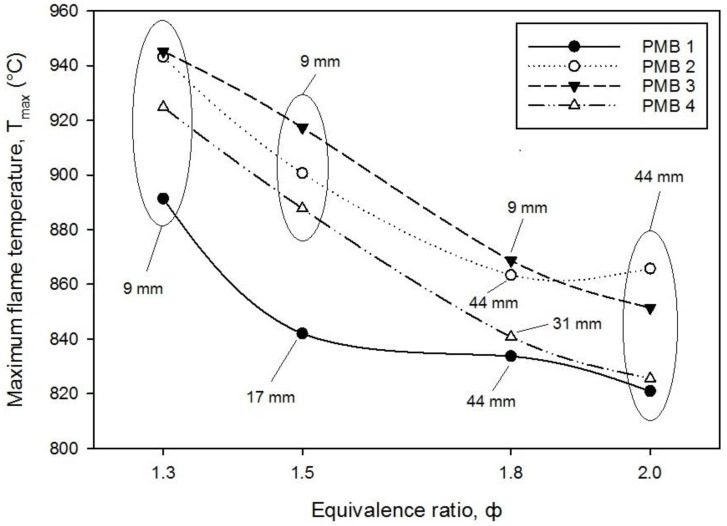
The maximum flame temperature and flame location for thermocouples height with varies equivalence ratio on various porous media configurations.

**Figure 6 entropy-22-01104-f006:**
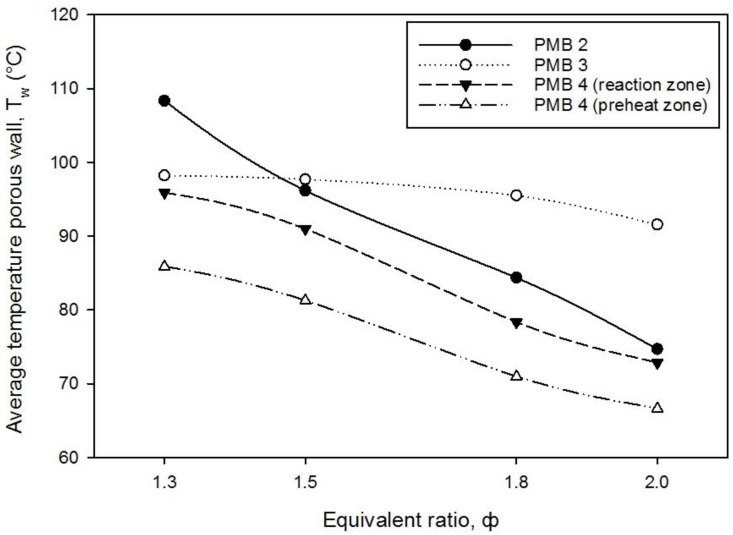
Average temperatures porous wall with varies equivalence ratios on various porous media configurations.

**Figure 7 entropy-22-01104-f007:**
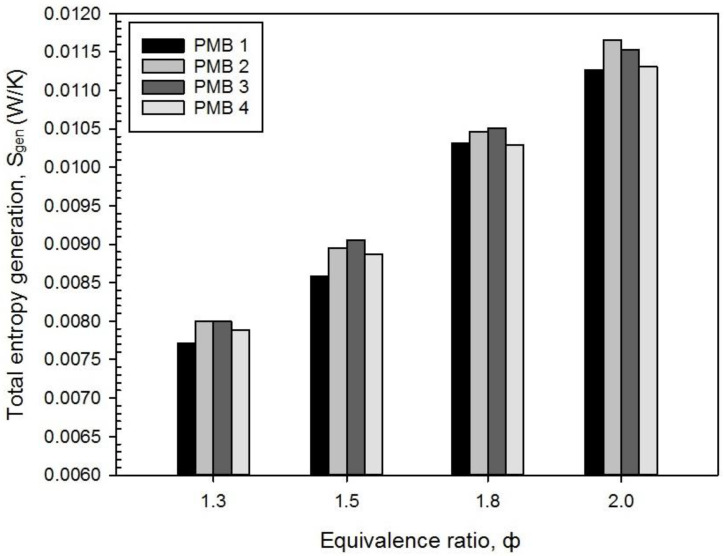
Total entropy generation, S_gen_ with variety of equivalence ratio on various porous media configurations.

**Figure 8 entropy-22-01104-f008:**
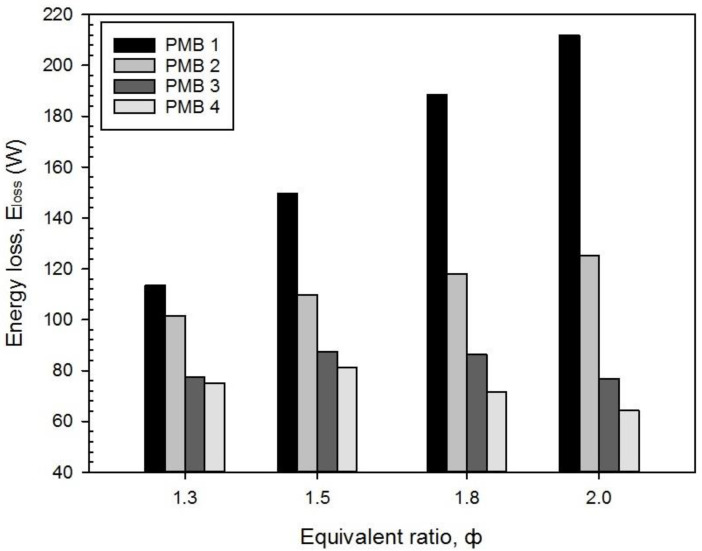
Energy loss, E_loss_ with varies equivalence ratio on various porous media configurations.

**Figure 9 entropy-22-01104-f009:**
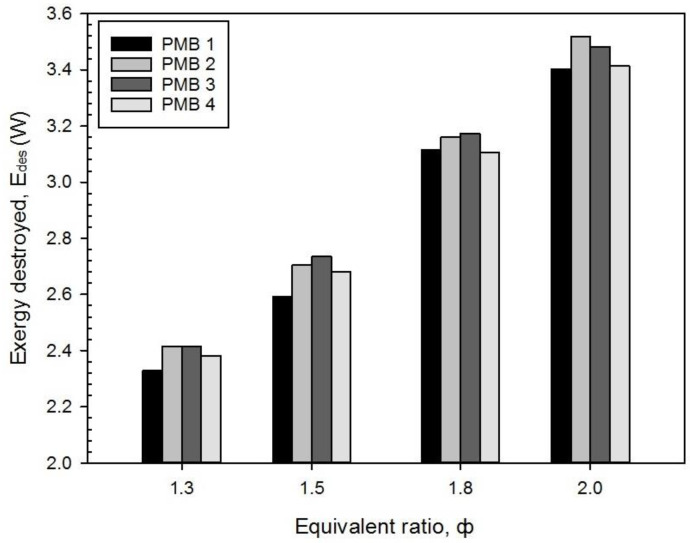
Exergy destroyed, E_des_ with varies equivalence ratio on various porous media configurations.

**Figure 10 entropy-22-01104-f010:**
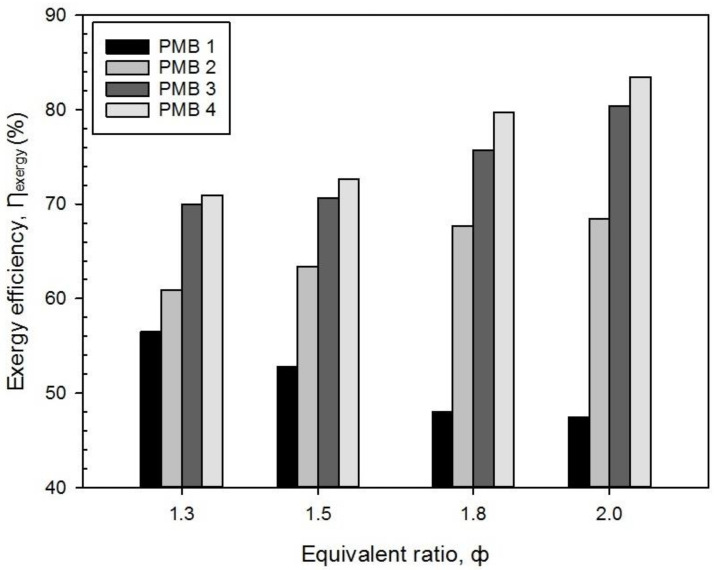
Exergy efficiency, ƞ_exergy_ with varies equivalence ratio on various porous media configurations.

**Figure 11 entropy-22-01104-f011:**
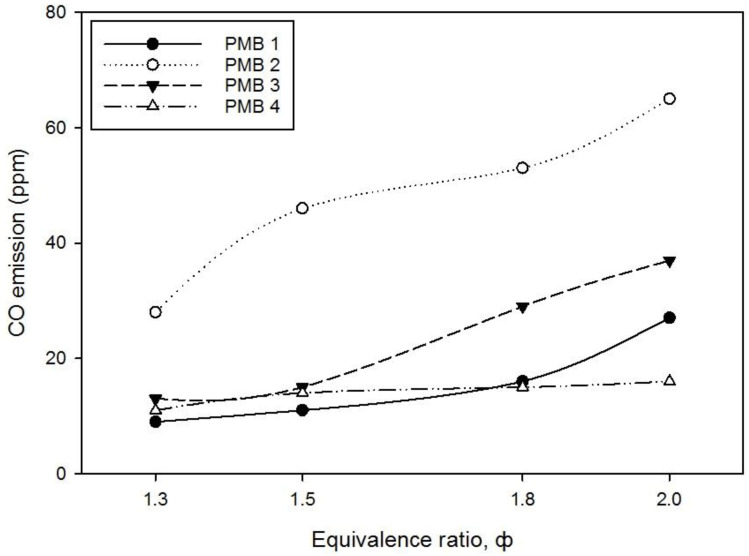
CO emission level (ppm) on various porous media configurations.

**Figure 12 entropy-22-01104-f012:**
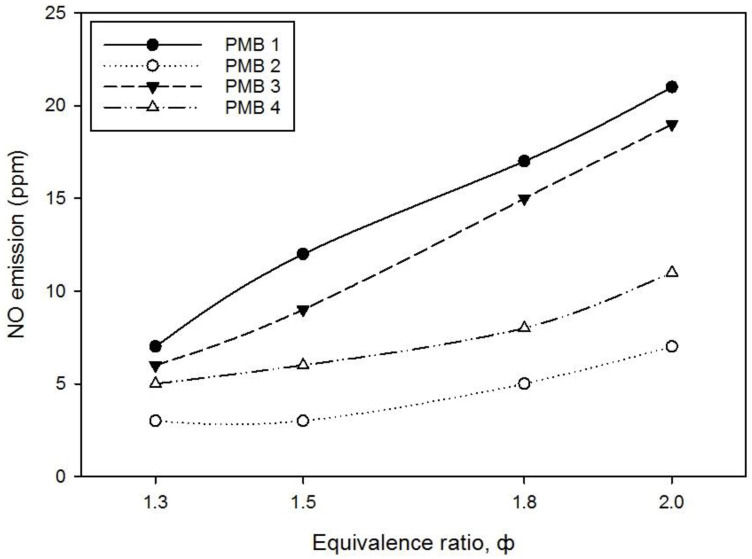
NO emission level (ppm) on various porous media configurations.

**Table 1 entropy-22-01104-t001:** Recent investigations of entropy generation rate using PMB systems.

Researchers	Type of Research	Fuel Mixture	Working Parameters	Main Outcomes
Jejurkar and Mishra [31]	3D Numerical	Hydrogen	Multi-step kinetics on annular shape combustor	The entropy generation rate increased from lean to rich mixtures; 0.5 ˂ φ ˃1.4.
Wenming et al. [32]	Numerical and Experimental	Hydrogen	Gap length of block insert	The combustor with a gap length of 4 mm produces the lowest entropy generation rate with higher gap length leads to the higher entropy generation.
Morsli et al. [33]	2D Numerical	Propane	Inlet velocity, oxygen percentage in air, and equivalence ratio	The contribution of thermal effect caused the effect in total generation of entropy.
Safer et al. [34]	Numerical	Hydrogen/carbon monoxide	Counter-flow flames of syngas mixtures	The total volumetric entropy generation decreases with H_2_ enrichment.
Mohammadi and Ajam [35]	2D Numerical	Methane	Multi-step mechanisms and variable porosity of porous media.	The entropy generation due to heat transfer has the highest contribution in entropy generation rate.
Zuo et al. [36]	3D Numerical	Hydrogen	Mass flow rate, equivalence ratio, materials, and inlet/outlet diameter ratios of variant diameter chambers.	The modified micro reactor has lower total entropy generation compared to the old micro reactor.
Ni et al. [37]	3D Numerical	Hydrogen	Axial location and height of geometric shape ribs	The chemical reaction and the conduction heat transfer contribute 70% and 15% of the total entropy generation.
Ansari and Amani [38]	3D Numerical	Methane	Flame stability, efficiencies, and emission on combined baffle-bluff	Combined baffle-bluff can reduce entropy generation rate by increase solid wall conductivity on both combustion and MTPV efficiency.
Wang et al. [39]	Numerical	Methane/hydrogen addition	Flow velocity, hydrogen addition in a micro-planar combustor	The entropy generation rate induced by chemical reaction, heat conduction and mass diffusion increase with the flow velocity.
Jiang et al. [40]	2D Numerical	Hydrogen	Flow velocity, fuel-air equivalence ratio, and effect heat recuperation	Higher flow velocity and H_2_/air equivalence ratio increase the rate of entropy generation.

**Table 2 entropy-22-01104-t002:** Porous media materials specifications for Alumina and Porcelain foam.

Specifications	Alumina	Porcelain
Type	Foam	Foam
Pore size	8 pores per centimeter (ppcm)	26 pores per centimeter (ppcm)
Porosity	84%	86%
Made by	Goodfellow Cambridge Limited (LS 3699006/1), England	School of Materials and Mineral Resources Engineering, Universiti Sains Malaysia

**Table 3 entropy-22-01104-t003:** Maximum uncertainty analysis with various PMB configurations.

x_i_		$\bar{x}$	$\sigma_{x}$	$\sigma_{\bar{x}}$	U_n_
Variables	Unit				
T_max_	°C	863.31	9.79	5.65	±24.33
T_w_	°C	74.49	4.13	2.38	±10.27
CO emission	ppm	28	3.61	2.08	±9
NO emission	ppm	12	2.00	1.15	±5

**Table 4 entropy-22-01104-t004:** Detailed values of mass flow rate for butane fuel and air at various equivalence ratios and flame velocities.

Equivalence Ratio, φ	1.3	1.5	1.8	2.0
Air (liters/min)	3.10			
Butane fuel (liters/min)	0.13	0.15	0.18	0.20
Flame velocity, S_L_ (m/s)	0.1296	0.1304	0.1316	0.1324

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
