# Peer review of "Entropy Generation and Exergy Analysis of Premixed Fuel-Air Combustion in Micro Porous Media Burner"

_entropy, 2020, doi:10.3390/e22101104_

Round 1

Reviewer 1 Report

Review of “Entropy generation and exergy analysis of premixed fuel-air combustion in micro porous media burner”

This paper analyzes experimentally a exergy analysis and entropy generation of premixed fuel-air combustion using single-and double-layers porous media burner. This research introduces a simple method of analysis to study the thermodynamics irreversibility in micro porous media combustion.

The work is interesting and the results are original. However, the paper should be revised before the possible acceptance to be published. Some comments and suggestions that the Authors should take into account are given in the following points:

1- In abstract: add more sentences for finding of the results.
2- English language needs improvement throughout the paper.
3- Nomenclature part should be revised, some of symbols are missed.
4- Novelty of the work should be highlighted.
5- Are you referring to some possible engineering applications? Please, discuss this point in the Introduction in more depth.
6- You should reference the papers that initiated the investigation of entropy study. See, for example;
- MHD Mixed Convection and Entropy Generation of Nanofluid in a Lid-Driven U-Shaped Cavity with Internal Heat and Partial Slip, Physics of Fluids, 31, 042006 (2019).
- Entropy Generation and Nanofluid Mixed Convection In A C-Shaped Cavity With Heat Corner and Inclined Magnetic Field, European Physical Journal Special Topics, 228(12) (2019) 2619-2645.
- , Inclined Magneto-Convection, Internal Heat and Entropy Generation of Nanofluid In An I-Shaped Cavity Saturated with Porous Media, Journal of Thermal Analysis and Calorimetry, (2020).

Author Response

The authors would like to thank you and all the reviewers for their detailed and constructive comments to further strengthen this manuscript. We are extremely grateful to all our reviewers for the effort and precious time put into the review of this manuscript. Each comment has been carefully considered and responded. The corresponding changes made in the revised manuscript are summarized below. The comments from all reviewers are in normal font and our responses are italicized.

Reviewer #1:

The work is interesting and the results are original. However, the paper should be revised before the possible acceptance to be published. Some comments and suggestions that the Authors should take into account are given in the following points:

1- In abstract: add more sentences for finding of the results.

Response:

Thank you for raising this point. We have reformulated and added salient findings in the abstract. The additional sentences are highlighted in red in the revised manuscript.

2- English language needs improvement throughout the paper.

Response:

After re-reading the manuscript with the reviewer’s comments in mind, we understand the concern with regards to the overall level of English language. We have gone through the manuscript carefully and the manuscript has been proofread.

3- Nomenclature part should be revised, some of symbols are missed.

Response:

We thank the reviewer for spotting the obvious errors in the manuscript. Some of the missing symbols have been added in the revised manuscript and they are highlighted in red.

4- Novelty of the work should be highlighted.

Response:

We deeply appreciate your suggestion and we found your comments extremely helpful. It is our intention, after all, to promote the novelty of our work that will help the readers’ understanding in this subject area. Accounting for the given suggestion, we have now included the novelty of our work in the conclusion section and the related contents are highlighted in red.   

5- Are you referring to some possible engineering applications? Please, discuss this point in the Introduction in more depth. 

Response:

Thanks for the constructive feedback and we apologize for not making our presentation clearer in the introduction section. There are numerous engineering applications of PMB and there are also notable evidences of PMB applications in emerging areas. We have now added your suggestions in the introduction section and the contents are highlighted in red. Accordingly, the added references are listed in the reference section (Refs. [4-8]).

6- You should reference the papers that initiated the investigation of entropy study. See, for example;
- MHD Mixed Convection and Entropy Generation of Nanofluid in a Lid-Driven U-Shaped Cavity with Internal Heat and Partial Slip, Physics of Fluids, 31, 042006 (2019).

- Entropy Generation and Nanofluid Mixed Convection In A C-Shaped Cavity With Heat Corner and Inclined Magnetic Field, European Physical Journal Special Topics, 228(12) (2019) 2619-2645.

- , Inclined Magneto-Convection, Internal Heat and Entropy Generation of Nanofluid In An I-Shaped Cavity Saturated with Porous Media, Journal of Thermal Analysis and Calorimetry, (2020).

Response:

We valued your suggestion with regards to these additional references. These aforementioned studies are now included in the introduction section and they are highlighted in red.

Reviewer 2 Report

Review of manuscript titled “Entropy generation and exergy analysis of premixed fuel-air combustion in micro porous media burner” by Ismail et al.

  1. Why is the porous media burner termed at “micro”, normal convention in combustion is that micro-scale means of the order of the flame thickness.
  2. In the essence, it should not be called porous media combustion as the flame is not stabilized inside the porous media. This clearly evident in the Figure 3 where you can see blue flame outside the burner which indicate the combustion. This mode of combustion is porous burner stabilized combustion.
  3. Technically, the exergy in the present work is done by taking into consideration energy transfer to water by heating (for specific configuration of vessel heating). What would happen if you use another configuration? Will the results be same?
  4. Does the author use any standards for the given configuration of vessel and water heating?
  5. How did the authors take into account the energy lost due to gases leaving the vessel without complete transfer of heat.
  6. Why is flame temperature is so low, 945 0C when adiabatic flame temperature for butane at phi = 1.3 would be about 1900 K (1627 0C).

Author Response

The authors would like to thank you and all the reviewers for their detailed and constructive comments to further strengthen this manuscript. We are extremely grateful to all our reviewers for the effort and precious time put into the review of this manuscript. Each comment has been carefully considered and responded. The corresponding changes made in the revised manuscript are summarized below. The comments from all reviewers are in normal font and our responses are italicized.

  1. Why is the porous media burner termed at “micro”, normal convention in combustion is that micro-scale means of the order of the flame thickness.

Response:

We thank the reviewer for pointing out the confusing term of ‘micro’ used throughout the original manuscript. The term ‘micro’ was derived based on the earlier investigation of Janvekar et al. (Ref. 43) who initially built the platform as a basis for our study. We understand that the term ‘micro’ might give an impression that the dimensional scale is in 10-6 m but for PMBs application the inner diameter of 25 mm falls under the category of a ‘micro’ burner. The term was also applied in the simulation work of Peng et al. (Ref. 26). Therefore, kindly note that the terminology ‘micro’ is referred to in the text.

  1. In the essence, it should not be called porous media combustion as the flame is not stabilized inside the porous media. This clearly evident in the Figure 3 where you can see blue flame outside the burner which indicate the combustion. This mode of combustion is porous burner stabilized combustion.

Response:

The reviewer is concerned with the terminology of porous media combustion used in the manuscript and we believe that the reviewer have raised a valid point. It is generally well understood among many researchers that porous media combustion is arguably a fairly loose term. To put the term in a more precise context, porous media combustion can actually be further classified into surface combustion and submerge combustion. Therefore, as correctly pointed by the reviewer, a stabilized mode of combustion can practically fall under one of these aforementioned categories. For clarity’s sake, we would like to draw the attention of the reviewer to the flame generated in our study, as shown in Fig. 3. It can be seen that the stabilized flame occurs at ф = 1.3 and the unstable flame occurs at ф = 2.0. The flame gradually developed from the blue flame on the surface of the porous medium to an orange-coloured flame when the value of ф was slowly increased. The flame for both conditions are surface combustion and the change in ф has an impact on the height as well as the physical shape of the flame. The explanation for the above mentioned observation can be found in the following paragraph in the manuscript:  

“From the observations, it can be seen that at ф = 1.3 the surface flame is stable and produces the blue flame for all PMB configurations. While, at ф = 2.0, the surface flame is unstable with flame front twisted and produce orange flame for all PMB configurations.”

Therefore, with regards to the reviewer’s concern, we are humbly requesting that the term porous media combustion which appears in the revised manuscript to be maintained in view of relevancy to the research.

  1. Technically, the exergy in the present work is done by taking into consideration energy transfer to water by heating (for specific configuration of vessel heating). What would happen if you use another configuration? Will the results be same?

Response:

We appreciate the reviewer’s concern and interest related to the heat transfer in the exergy context. We interpreted your comments as arising from our shortcomings in explaining the water boiling test (WBT) which was performed in this study. WBT was performed to determine the heat transfer that took place which in turn affect the exergy content. The test is a standard procedure and the procedure is elucidated in great detail in other research works (See for example: Patangi et al. 2011, Ismail et al. 2013, and Mustafa et al. 2015). Given that the equation used (Eq. 5) is a function of specific heats and mass of the water and the container, the results will quantitatively be the same even if other configurations are used to perform the test.

Ismail AK, Abdullah MZ, Zubair M, Ahmad ZA, Jamaludin AR, Mustafa KF, Abdullah MN. Application of porous medium burner with micro cogeneration system. Energ 2013;50:131-42.

Patangi VK, Mishra SC, Muthukumar P, Reddy R. Studies on porous radiant burners for LPG (liquefied petroleum gas) cooking applications. Energ 2011;36:6074-80.

Mustafa KF, Abdullah S, Abdullah MZ, Sopian K. Experimental analysis of a porous burner operating on kerosene-vegetable cooking oil blends for thermophotovoltaic power generation. Energy Convers Manage 2015;96:544-60.

  1. Does the author use any standards for the given configuration of vessel and water heating?

Response:

We believe the issue raised by the reviewer is interrelated to the point discussed in #3 above. The procedure adopted in the study has been explained and additional information can also be found in Refs. (43−44).

  1. How did the authors take into account the energy lost due to gases leaving the vessel without complete transfer of heat.

Response:

We appreciate the comments by the reviewer on the determination of the energy lost. We would like to state that the energy loss due to the gases leaving the vessel was calculated based on the energy carried through the exhaust pipe, and this calculation is shown in Eq. (6) in the manuscript. The equation used the difference between the energy supplied from butane and the energy generated from the combustion process. 

  1. Why is flame temperature is so low, 945 0C when adiabatic flame temperature for butane at phi = 1.3 would be about 1900 K (1627 0C).

Response:

The reviewer raised the point on a huge temperature difference between the adiabatic flame and the experimentally measured value. We agree that we should clarify further that the adiabatic flame temperature is based on the theoretical calculation. As rightly point out by the reviewer, the adiabatic flame temperature of butane is 1627 °C and this is significantly higher that the measured value in our study, which was only 945 °C. The theoretically calculated adiabatic flame temperature hinged on the primary assumption that there is zero energy loss in a close system and resulting in a perfect combustion. However, the experimentally measured will be subjected to a radiant loss to the ambient. In addition, although the porous media combustion is widely known as excess enthalpy combustion which means higher temperature, this combustion is referred to a submerged combustion. The temperature generated in the submerge combustion tends to be very high but the same cannot be said for a surface combustion.   

Reviewer 3 Report

This paper investigated the exergy efficiency and entropy generation of premixed fuel-air combustion. Porous medias of different structures were used in the burner. Although some meaningful conclusions have been obtained, some parts about the methodology and the explanations were unclear. More clarifications and better explanations should be provided before publication.

(1) PMB4 designed two-layer porous structure with a preheating zone and a combustion zone, which is similar to the references in the second paragraph. However, the results showed that the flame was not stabilized in the porous media of the combustion zone, but on the surface of the porous media. Porous media burner usually refers to the burner using porous media combustion technology. So the content of this paper is inconsistent with the title and some content in the introduction.

(2) “The experiments were carried out in a rich fuel condition with equivalence ratio, ф range from 1.3 to 2.0”. Why all experimental conditions are fuel-rich? The authors should give an explanation for this choice.

(3) Some experimental details are missing in the experimental setup part of this paper. For example, the experimental setup of measuring the temperature of the burner wall is not introduced, but the results of the wall temperature are discussed in the results section. The location of the exhaust gas sampling is not described.

(4) Why is the maximum flame temperature measured in the experiment much lower than the combustion temperature of butane? Is there a radiant heat loss correction for the temperature measured by the thermocouple?

(5) Figure 4 shows the distance of the flame from the porous surface. The Flow rate and velocity have a great influence on the lift-off distance, so they should be provided when introducing the experimental conditions.

(6) Page 15, “PMB2 shows the lowest NOx emissions for all equivalence ratios. In other words, the biggest pore size increased the flame velocity, thus reduced NOx emissions”. The effect of pore size on the flame velocity is mentioned here. How is this rule derived? Is it based on the current experimental results or previous research?

(7) As the equivalence ratio increases, the maximum flame temperature decreases, but NOx emission increases. Why does NOx emission not decrease with the reduction of flame temperature?

Author Response

Please see the attachment for the reviewer responses and revised manuscript.

Round 2

Reviewer 2 Report

The authors have addressed all my queries properly and improved the manuscript appropriately. This manuscript can be accepted in the present form.